# A Dynamic Self-Tuning Maximum Correntropy Kalman Filter for Wireless Sensors Networks Positioning Systems

**Tianrui Liao** , **Kaoru Hirota, Xiangdong Wu** , **Shuai Shao and Yaping Dai \***

School of Automation, Beijing Institute of Technology, Beijing 100081, China
\* Correspondence: daiyaping@bit.edu.cn; Tel.: +86-010-6891-5616

**Abstract:** To improve the accuracy of the maximum correntropy Kalman filter (MCKF) in wireless sensors networks (WSNs) positioning, a dynamic self-tuning maximum correntropy Kalman filter (DSTMCKF) is proposed, where innovation and the sensors information of the WSNs are used to adjust the noise covariance matrices, and the maximum correntropy criterion is the criterion for the filter's optimality. By dynamically adjusting the noise covariance matrices, the DSTMCKF ensures that the correntropy distribution is accurate in the presence of non-Gaussian noise (NGN), thus improving its ability to handle the NGN. In simulation and real environment positioning experiments, the DSTMCKF is used to compare with the MCKF, variable kernel width–maximum correntropy Kalman filter (VKW-MCKF) and robust minimum error entropy Kalman filter (R-MEEKF). Among the four filters, the DSTMCKF has the highest accuracy, and the error of the DSTMCKF is reduced by 34.5, 42.9 and 40.0%, respectively, compared with the MCKF, VKW-MCKF and R-MEEKF in the real-world environment positioning experiment. The application of the DSTMCKF in WSNs positioning systems improves the stability of the control systems because of the rising positioning accuracy, which makes WSNs positioning systems more widely used in scenarios requiring high stability, such as automatic parking.

**Keywords:** wireless sensors networks; maximum correntropy Kalman filter; dynamic self-tuning algorithm; range-based positioning

## 1. Introduction

With the development of sensors technology, wireless sensors networks (WSNs) positioning systems are widely used in factories, docks, warehouses and other places and have broad development prospects [1–3]. In recent years, Gulo et al. [4] and Duan et al. [5] have applied WSNs positioning technology to occasions other than industrial applications, such as mobile phone positioning and vehicle positioning, further improving the practicability of WSNs positioning technology. Because of a low cost and high positioning accuracy, the range-based positioning method has become the most shared WSNs positioning method. Authors such as Xiong et al. [6], Na et al. [7] and Nguyen et al. [8] have all adopted range-based positioning methods to effectively improve the positioning accuracy.

However, the range-based positioning algorithm is seriously affected by non-Gaussian noise (NGN) [9,10], which results in its inaccuracy in practical application. Huang et al. [11], Fang et al. [12] and Navon et al. [13] all pointed out that WSNs positioning systems are affected by various NGNs, which seriously reduce the positioning accuracy.

To suppress the influence of NGN, various robust Kalman filter methods are applied to WSNs positioning systems [5,14]. The classical robust Kalman filter includes the H∞ filter [15,16] and the strong tracking Kalman filter [17,18]. However, these filters are not effective in dealing with the noise in WSNs positioning problems, and the positioning accuracy decreases seriously.

In recent years, the maximum correntropy Kalman filter (MCKF) was proposed by Chen et al. [19], according to the information theoretic learning criterion. Compared with

the traditional robust Kalman filter, the MCKF has a higher accuracy and has the ability to deal with various NGNs [20]. Owing to its excellent performance, the MCKF is rapidly applied to various systems [21,22]. However, in WSNs positioning systems, most of the NGN comes from the electromagnetic radiation of electronic equipment and non-line-of-sight signals due to occlusion and refraction. These noises are usually much larger than the Gaussian white noise in the channel, and the size and duration are difficult to predict. With the MCKF, it is hard to deal with these noises, so the filtering accuracy of the MCKF will be reduced in the WSNs positioning systems [23,24].

To improve the performance of the MCKF in WSNs positioning systems, a dynamic self-tuning maximum correntropy Kalman filter (DSTMCKF) is proposed. In order to deal with the non-Gaussian noise, a dynamic self-tuning (DST) algorithm is proposed to combine with the MCKF. The DST algorithm uses innovation and the information from sensors to modify the noise covariance matrices, and the MCKF uses a maximum correntropy criterion (MCC) for the filter's optimality.

By using the DST algorithm to adjust the noise covariance matrices in real time, the response speed and accuracy of the filter when encountering non-Gaussian noise are improved. Compared with the minimum mean squared error (MMSE) criterion, the maximum correntropy criterion has stronger robustness to ensure the convergence of the filter when encountering non-Gaussian noise. By combining the DST algorithm and the MCC, the DSTMCKF ensures that the correntropy distribution is accurate in the presence of NGN, thus improving its ability to handle NGN.

To verify the performance of the DSTMCKF, three filters, including the MCKF, variable kernel width–maximum correntropy Kalman filter (VKW-MCKF) and robust minimum error entropy Kalman filter (R-MEEKF), are used to compare with the DSTMCKF in the simulation experiments and real-world dynamic positioning experiments. The simulation experiments consider the different non-Gaussian noise in the motion state and observation, and the real-world dynamic positioning experiment is selected in the real office scene.

The rest of the paper is organized as follows: In Section 1, the principle of the MCKF is presented in detail, along with the limitations of the MCKF. In Section 2, a DSTMCKF is proposed, and the principle of the DSTMCKF is described. In Section 3, the simulation and real-world dynamic experiments results are presented, both comparing the DSTMCKF to other filters and exploring some of its qualities.

## 2. Maximum Correntropy Kalman Filter

### 2.1. Principle of MCKF

To suppress the influence of NGN on positioning accuracy, the MCKF is used to deal with NGN and has achieved promising results. Compared with the MMSE, the MCC has a stronger robustness and lower risk when facing NGN [25]. Therefore, compared with the classical Kalman filter, the MCKF is more suitable for dealing with NGN [19].

The MCC is an optimal estimation criterion, also known as the maximum information criterion. For two variables $X$ and $Y$ whose joint distribution function is $F_{XY}(x, y)$, the definition of correntropy $V(X, Y)$ is

$$V(X, Y) = E[\kappa(X, Y)] = \int \kappa(X, Y) dF_{XY}(x, y), \tag{1}$$

where $E$ is the expectation operator and $\kappa(X, Y)$ represents the kernel function of correntropy. Generally, $\kappa(X, Y)$ is

$$\kappa(X, Y) = G_\sigma(e) = \exp(-\frac{e^2}{2\sigma^2}), \tag{2}$$

where $G_\sigma$ is the Gaussian function, $e$ is equal to $X - Y$ and $\sigma$ is the bandwidth of the Gaussian kernel function.

When applying MCC to KF, a linear system is considered first. The state equation and observation equation of linear system are

$$\begin{cases} \mathbf{x}_k = \mathbf{F}_k \mathbf{x}_{k-1} + \mathbf{w}_k \\ \mathbf{y}_k = \mathbf{H}_k \mathbf{x}_k + \mathbf{v}_k \end{cases}, \tag{3}$$

where $\mathbf{x}_k \in \Re^m$ is the system state at the $k$-th sampling, $\mathbf{w}_k$ is the process noise, $\mathbf{y}_k \in \Re^n$ is the observation at the $k$-th sampling and $\mathbf{v}_k$ is the observation noise. State transition matrix and observation matrix are $\mathbf{F}_k \in \Re^{m \times m}$ and $\mathbf{H}_k \in \Re^{m \times n}$ respectively.

Then, the process noise covariance matrix and observation noise covariance matrices $\mathbf{Q}_k$ and $\mathbf{R}_k$ are

$$\begin{cases} \mathbf{Q}_k = E\left(\mathbf{w}_k \mathbf{w}_k^{\mathrm{T}}\right) \\ \mathbf{R}_k = E\left(\mathbf{v}_k \mathbf{v}_k^{\mathrm{T}}\right) \end{cases}. \tag{4}$$

In order to apply the MCC to Kalman filter, it is necessary to establish the correntropy model. The a priori estimate $\mathbf{D}_k$ and a posteriori estimate $\mathbf{W}_k \hat{\mathbf{x}}_{k|k}$ in the Kalman filter are used as correlation variables in the correntropy model. Then, construct matrices $\mathbf{D}_k$ and $\mathbf{W}_k$ are

$$\mathbf{B}_k = \mathbf{chol}\left( \begin{bmatrix} \mathbf{P}_{k|k-1} & 0 \\ 0 & \mathbf{R}_k \end{bmatrix} \right), \tag{5}$$

$$\mathbf{D}_k = \mathbf{B}_k^{-1} [\; \hat{\mathbf{x}}_{k|k-1} \quad \mathbf{y}_k \;]^{\mathrm{T}}, \tag{6}$$

$$\mathbf{W}_k = \mathbf{B}_k^{-1} [\; I \quad \mathbf{H}_k \;]^{\mathrm{T}}, \tag{7}$$

where $\mathbf{P}_{k|k-1}$ is the a priori estimate of process noise covariance matrix. The prior estimation of the process noise covariance matrix $\mathbf{P}_{k|k-1}$ in this iteration can be calculated by using the state transition matrix $\mathbf{F}_k$ and the posterior estimation of the process noise covariance matrix $\mathbf{P}_{k-1}$ obtained in the previous iteration. In addition, $\mathbf{chol}(x)$ is a Cholesky decomposition operator, which means Cholesky decomposition of $x$.

In order to calculate the correntropy of $\mathbf{D}_k$ and $\mathbf{W}_k \hat{\mathbf{x}}_{k|k}$, $\mathbf{E}_k$ needs to be calculated as

$$\mathbf{E}_k = \mathbf{D}_k - \mathbf{W}_k \hat{\mathbf{x}}_{k|k}, \tag{8}$$

where $\hat{\mathbf{x}}_{k|k}$ is the a priori estimate of the $k$-th sampling.

Then, the correntropy $V(\mathbf{D}_k, \mathbf{W}_k \hat{\mathbf{x}}_{k|k})$ of $\mathbf{D}_k$ and $\mathbf{W}_k \hat{\mathbf{x}}_{k|k}$ is

$$V(\mathbf{D}_k, \mathbf{W}_k \hat{\mathbf{x}}_{k|k}) = \frac{1}{m+n} \sum_{i=1}^{m+n} G_\sigma(\tilde{\mathbf{e}}_k(i)), \tag{9}$$

where $\mathbf{e}_k(i)$ is the $i$-th element of vector $\mathbf{E}_k$.

According to MCC, when $V(\mathbf{D}_k, \mathbf{W}_k \hat{\mathbf{x}}_{k|k})$ is the maximum, $\hat{\mathbf{x}}_{k|k}$ is the optimal value. To calculate the optimal value of $\hat{\mathbf{x}}_{k|k}$, fixed-point iteration method is used to calculate $\hat{\mathbf{x}}_{k|k}$. The steps of MCKF are summarized as follows:

(a) Prediction:

$$\begin{cases} \hat{\mathbf{x}}_{k|k-1} = \mathbf{F}_k \hat{\mathbf{x}}_{k-1|k-1} \\ \mathbf{P}_{k|k-1} = \mathbf{F}_k \mathbf{P}_{k-1} \mathbf{F}_k^{\mathrm{T}} + \mathbf{Q}_{k-1} \end{cases}, \tag{10}$$

where $\mathbf{x}_{k|k-1}$ is the prior estimate of the current iteration obtained by the state transition matrix $\mathbf{F}_k$ and the state posterior estimation $\mathbf{x}_{k-1|k-1}$ of the previous iteration.

(b) Update required parameters of MCKF:

$$\begin{cases} \mathbf{B}_p = \mathbf{chol}(\mathbf{P}_{k|k-1}) \\ \mathbf{B}_r = \mathbf{chol}(\mathbf{R}_k) \end{cases}. \tag{11}$$

(c) Fixed-point iteration:

$$\begin{cases} \tilde{\mathbf{C}}_x = \text{diag}(G_\sigma(\tilde{\mathbf{e}}_k(1)), \cdots, G_\sigma(\tilde{\mathbf{e}}_k(n))) \\ \tilde{\mathbf{C}}_y = \text{diag}(G_\sigma(\tilde{\mathbf{e}}_k(n+1)), \cdots, G_\sigma(\tilde{\mathbf{e}}_k(n+m))) \end{cases}, \tag{12}$$

$$\begin{cases} \tilde{\mathbf{R}}_k = \mathbf{B}_r \tilde{\mathbf{C}}_y^{-1} \mathbf{B}_r^T \\ \tilde{\mathbf{P}}_{k|k-1} = \mathbf{B}_p \tilde{\mathbf{C}}_x^{-1} \mathbf{B}_p^T \end{cases}, \tag{13}$$

$$\tilde{\mathbf{K}}_k = \tilde{\mathbf{P}}_{k|k-1} \mathbf{H}^T_k [\mathbf{H}_k \tilde{P}_{k|k-1} \mathbf{H}_k^T + \tilde{\mathbf{R}}_k]^{-1}, \tag{14}$$

$$\hat{\mathbf{x}}_{k|k}(t) = \hat{\mathbf{x}}_{k|k-1} + \tilde{\mathbf{K}}_k [\mathbf{y}_k - \mathbf{H}_k \hat{\mathbf{x}}_{k|k-1}], \tag{15}$$

where $\tilde{\mathbf{K}}_k$ is the Kalman filter gain, and $t$ represents the number of iteration rounds of fixed-point iteration.

(d) The judgement criterion for iteration stop is

$$\frac{\left\| \hat{\mathbf{x}}_{k|k}(t) - \hat{\mathbf{x}}_{k|k}(t-1) \right\|}{\left\| \hat{\mathbf{x}}_{k|k}(t) \right\|} \le \varepsilon, \tag{16}$$

where $\varepsilon$ is the error threshold. It can be seen from document [19], to ensure the accuracy and numerical stability of the algorithm, the value of $\varepsilon$ is $10^{-4}$ in this paper. When Equation (16) holds, the fixed-point iteration is stopped; otherwise, continue with step (c).

(e) Update the filtering results and prepare for the next iteration:

$$\hat{\mathbf{x}}_{k|k} = \hat{\mathbf{x}}_{k|k}(t), \tag{17}$$

$$\mathbf{P}_k = [I - \tilde{\mathbf{K}}_k \mathbf{H}_k] \mathbf{P}_{k|k-1} [I - \tilde{\mathbf{K}}_k \mathbf{H}_k]^{\mathrm{T}}. \tag{18}$$

Through the above steps, the state with maximum correntropy is estimated. MCKF replaces MMSE criteria with MCC, which has higher robustness when the noise size cannot be determined. However, MCKF has limitations in practical application, which result in underdeveloped filtering effect and even divergence of filtering results.

### 2.2. Limitations of MCKF

In practical application, when the noise type or target motion state is multiple, the filtering accuracy of the MCKF will decrease to a certain extent. In order to explain this phenomenon, a situation is considered when the MCKF is working properly [26].

Considering that the a priori estimate and observation measurement are only one dimension. Supposing the state prior estimate $\mathbf{x}_{k|k-1}$ is 40, the state observation value $\mathbf{y}_k$ is 60, process noise covariance matrix $\mathbf{P}_{k|k-1}$ is 6, observation noise covariance matrix $\mathbf{R}_k$ is 1 and the value of Gaussian kernel $\sigma$ is 5. Where $\mathbf{P}_{k|k-1}$ and $\mathbf{x}_{k|k-1}$ are the prior estimators of the current iteration obtained by the state transition matrix $\mathbf{F}_k$ and the posterior estimators of the previous iteration, and the values are not affected by the observation noise of the current iteration.

If the values of the estimated noise covariance matrices $\mathbf{P}_{k|k-1}$ and $\mathbf{R}_k$ are not much different from the values of the noise covariance in reality, the distribution of correntropy is shown in Figure 1a. The correntropy has only one maximum value; hence, the fixed-point iteration ensures convergence to the point with maximum correntropy. In this case, MCKF has strong robustness and has a better ability to deal with NGN.

On the contrary, when the MCKF is interfered by noise of an unpredictable size, the actual observed noise covariance is inconsistent with $\mathbf{R}_k$. At this time, the observation $\mathbf{y}_k$ will also change due to the influence of observation noise. If the noise further increases, the difference between state a priori estimation and state observation will further increase.

Assuming that the state view measurement is 80 and other variables remain unchanged, the distribution of correntropy is shown in Figure 1b.

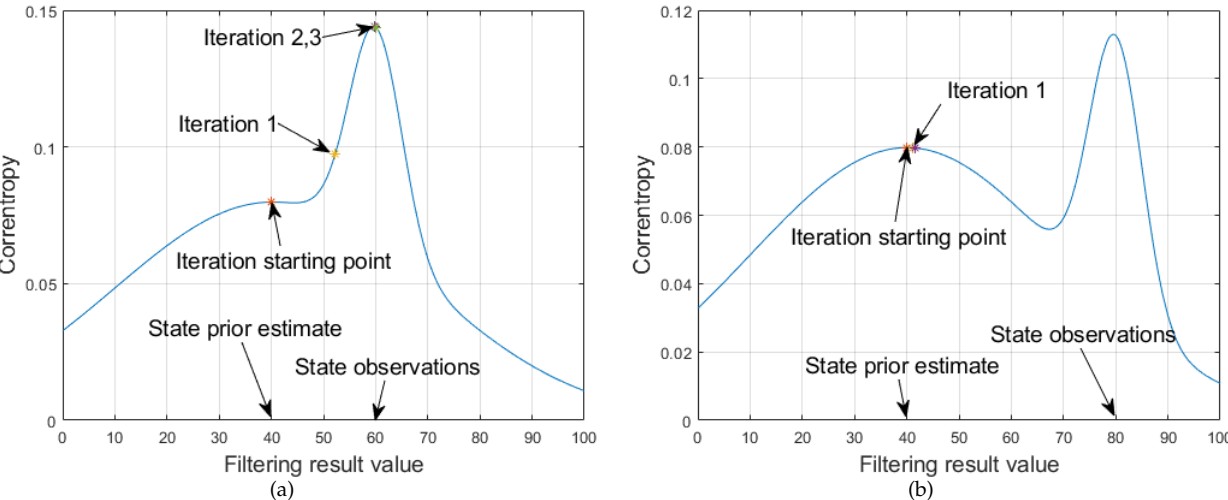

**Figure 1.** Distribution of correntropy in different scenes. (**a**) Distribution of correntropy when parameters are accurate; (**b**) distribution of correntropy when NGN increases.

There are three extreme points in the solution space of the fixed-point iteration algorithm under these conditions, which do not satisfy the iteration convergence condition and provoke a large error in the calculation results. Thus, when the observation noise or the system motion state is multiple, the performance of the MCKF is underdeveloped.

One reason behind this phenomenon is that the MCKF is sensitive to the values of $\mathbf{P}_{k|k-1}$ and $\mathbf{R}_k$, but the values cannot be dynamically corrected. When the values of $\mathbf{P}_{k|k-1}$ and $\mathbf{R}_k$ are inconsistent with the reality, there will be errors in the distribution of correntropy, resulting in the degradation of the MCKF performance.

Another reason is that the fixed-point iteration is difficult to converge to the maximum correntropy point when the observation noise or the system motion state is multiple. Under these conditions, there are multiple extreme points in the solution space, which do not meet the convergence condition of the fixed-point iteration. The fixed-point iteration method may converge to the extreme point far away from the maximum correntropy point, resulting in a large error.

To solve this problem, it is necessary to modify $\mathbf{P}_{k|k-1}$ and $\mathbf{R}_k$ in each iteration [11,27]. The adaptive algorithms realize the dynamic correction of parameters to a certain extent, and yet when dealing with NGN in WSNs positioning systems, the adaptive algorithm has a slow response speed and insufficient robustness. Therefore, several algorithms have been proposed to improve the filtering accuracy of the MCKF in recent years.

*2.3. Improved Algorithm of MCKF*

With the aim of improving the performance of the MCKF, several algorithms have been proposed in recent years. There are two common ways to improve the accuracy: one is to adjust the Gaussian kernel bandwidth and the other is to optimize the maximum correntropy criterion.

Much of the research has examined how to improve the filtering precision of the MCKF by using a modified Gaussian kernel bandwidth, and the variable kernel width–maximum correntropy Kalman filter (VKW-MCKF) proposed by Huang et al. [24] has achieved positive results. The adaptive algorithm is used to adjust the Gaussian kernel bandwidth in the VKW-MCKF, so that the Gaussian kernel bandwidth can be adjusted adaptively with the size of the noise, which effectively improves the accuracy and robustness of the MCKF. However, the VKW-MCKF does not have the ability to adjust the noise covariance matrix,

so there is still an inaccurate distribution of the correntropy when the noise does not match the estimation.

In recent years, a series of studies have expected to improve the filtering accuracy of the MCKF by optimizing the maximum correntropy criterion, which the robust minimum error entropy Kalman filter (R-MEEKF) proposed by Chen et al. [23] performed favorably. The R-MEEKF optimizes the maximum correntropy principle to the minimum entropy principle, which further improves the ability of the filter to process non-Gaussian noise. Nonetheless, the R-MEEKF algorithm requires a trade-off between the convergence speed and steady-state error, and it is often difficult to guarantee that both performances meet the requirements.

In summary, although several algorithms have improved the filtering accuracy of the MCKF in recent years, none of the above improved algorithms can solve the problem that the MCKF is difficult to handle the variable process and observation noise. The above improved algorithms are difficult to apply in positioning systems with complex and variable noise; hence, a dynamic self-tuning algorithm is proposed to adjust the values of the noise covariance matrix $\mathbf{P}_{k|k-1}$ and $\mathbf{R}_k$.

## 3. Dynamic Self-Tuning Maximum Correntropy Kalman Filter

### 3.1. Determination of Noise Type for WSNs Positioning

In order to dynamically adjust $\mathbf{P}_{k|k-1}$ or $\mathbf{R}_k$, it is necessary to judge whether NGN appears in the observation. A WSNs positioning quality index is proposed, which is calculated according to the distance information between the tag and anchors.

The quality index is obtained by trilateral positioning method, and the principle is shown in Figure 2. In Figure 2a, $(X_1, Y_1)$, $(X_2, Y_2)$ and $(X_3, Y_3)$ are the coordinates of the three anchors. $d_1$, $d_2$ and $d_3$ are the distances to tag measured by the three anchors [28,29]. Then, the coordinate $(X_T, Y_T)$ of tag is

$$
\begin{bmatrix} X_T \\ Y_T \end{bmatrix} = \frac{1}{2} \begin{bmatrix} X_B - X_A & Y_B - Y_A \\ X_C - X_A & Y_C - Y_A \end{bmatrix}^{-1}
$$
$$
\cdot \begin{bmatrix} d_A^2 - d_B^2 + X_B^2 - X_A^2 + Y_B^2 - Y_A^2 \\ d_A^2 - d_C^2 + X_C^2 - X_A^2 + Y_C^2 - Y_A^2 \end{bmatrix}.
\tag{19}
$$

As shown in Figure 2b, when there is an error in the measured distance, the three circles in the triangulation positioning method cannot intersect at the same point. At this time, the distance $\hat{d}_i$ from the positioning result $(X_T, Y_T)$ to the Anchor $i$ is not equal to the measured distance $d_i$. Utilizing the difference between $\hat{d}_i$ and $d_i$, a WSNs positioning quality evaluation index is proposed.

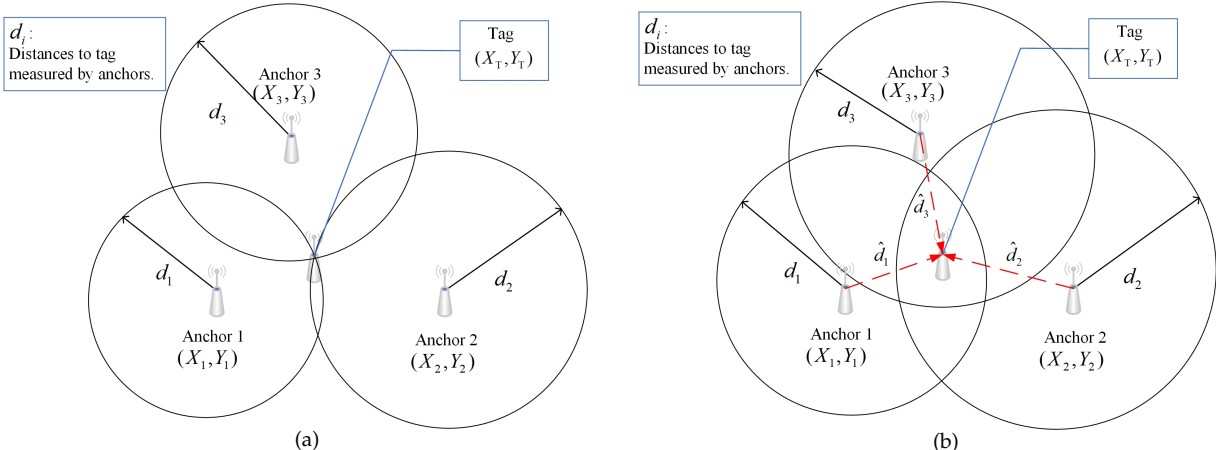

(a)　　　　　　　　　　　　　　　　　　　　　　　　(b)

**Figure 2.** Different quality positioning result. (**a**) High-quality positioning result; (**b**) low-quality positioning result.

The positioning quality index $L$ is

$$L = \frac{\sum_{i=1}^{N}\left|d_i - \hat{d}_i\right|}{N},$$

(20)

where $N$ is the number of anchor used for positioning.

The $L$ obtained by the $k$-th sampling is denoted as $L_k$. When the $L_k$ is large, it indicates that the difference between $\hat{d}_i$ and $d_i$ is large in this sampling. In addition, the greater the difference between $\hat{d}_i$ and $d_i$, the less reliable the positioning result. At this time, NGN appears in the observation, and $\mathbf{R}_k$ needs to be adjusted.

From another perspective, if $(X_T, Y_T)$ is taken as the unbiased estimation of the positioning result, $L_k$ is the mean ranging error of each positioning result. To test whether the ranging error satisfies the hypothesis, the hypothesis testing method is used to determine whether $L_k$ is within the rejection domain. When $L_k$ is not in the rejection domain, it means that the ranging error conforms to the hypothesis; otherwise, it means that the ranging error does not conform to the hypothesis. The critical point of the rejection domain is the threshold $L_{thr}$, so when $L_k$ is greater than $L_{thr}$, it is considered that the ranging error does not conform to the hypothesis, and there is NGN in the observation.

### 3.2. Dynamic Self-Tuning Algorithm

After determining whether the NGN exists in the observation, the next step is to correct the noise covariance matrices. A dynamic self-tuning (DST) algorithm is presented, which uses innovation to adjust the value of $\mathbf{P}_{k|k-1}$ or $\mathbf{R}_k$. Record innovation $\mathbf{z}_k$ as:

$$\mathbf{z}_k = \mathbf{y}_k - \mathbf{H}_k\mathbf{x}_{k|k-1}.$$

(21)

When the estimation of the model is accurate and the noise meets the Gaussian distribution, the expectation $\mathbf{C}_k$ of innovation covariance is shown in Equation (22):

$$\mathbf{C}_k \overset{\Delta}{=} \mathbf{H}_k\mathbf{P}_{k|k-1}\mathbf{H}_k^{\mathrm{T}} + \mathbf{R}_k.$$

(22)

In the actual filtering process, the covariance $\tilde{\mathbf{C}}_k$ is calculated in Equation (23). In addition, according to [30], after weighing the robustness of the algorithm and the response speed to NGN, the forgetting factor $\lambda$ is taken as 0.95.

$$\tilde{\mathbf{C}}_k = \begin{cases} \mathbf{z}_k\mathbf{z}_k^{\mathrm{T}}, & k = 1 \\ \frac{\lambda\tilde{\mathbf{C}}_{k-1}+\mathbf{z}_k\mathbf{z}_k^{\mathrm{T}}}{1+\lambda}, & k > 1 \end{cases}.$$

(23)

When the estimation of the model is accurate and the noise meets the Gaussian distribution, the value of $\tilde{\mathbf{C}}_k$ is similar to $\mathbf{C}_k$. On the contrary, when the motion state changes or there is NGN in the observation, the difference between $\tilde{\mathbf{C}}_k$ and $\mathbf{C}_k$ will increase significantly. At this time, $\mathbf{P}_{k|k-1}$ or $\mathbf{R}_k$ needs to be tuned to force $\tilde{\mathbf{C}}_k$ equal to $\mathbf{C}_k$.

If the $L_k$ or $L_{k-1}$ in Section 3.1 has an increased value than that of the threshold $L_{thr}$, there is NGN in the observation, so it is necessary to adjust $\mathbf{R}_k$ to $\hat{\mathbf{R}}_k$, forcing Equation (24) to hold.

$$\tilde{\mathbf{C}}_k = \mathbf{H}_k\mathbf{P}_{k|k-1}\mathbf{H}_k^{\mathrm{T}} + \hat{\mathbf{R}}_k.$$

(24)

Note that the intermediate variable $\mathbf{N}_k$ is:

$$\mathbf{N}_k = \tilde{\mathbf{C}}_k - \mathbf{H}_k\mathbf{P}_{k|k-1}\mathbf{H}_k^{\mathrm{T}}.$$

(25)

Equation (24) holds when $\hat{\mathbf{R}}_k$ is equal to $\mathbf{N}_k$. In order to ensure that $\hat{\mathbf{R}}_k$ is positive definite, the diagonal element of $\mathbf{R}_k$ is set by the expansion coefficient $\beta_{i|k}$, and then

the value of non-diagonal element is corrected by using the correlation coefficient, as shown below:

$$\beta_{i|k} = \max[1, \frac{n_{ii}}{r_{ii}}], \qquad (26)$$

$$\hat{\mathbf{R}}_k = \begin{bmatrix} \beta_{1|k}r_{11} & \sqrt{\beta_{1|k}\beta_{2|k}}r_{12} & \cdots & \sqrt{\beta_{1|k}\beta_{n|k}}r_{1n} \\ \sqrt{\beta_{1|k}\beta_{2|k}}r_{21} & \beta_{2|k}r_{22} & \cdots & \sqrt{\beta_{2|k}\beta_{n|k}}r_{2n} \\ \vdots & \vdots & \ddots & \vdots \\ \sqrt{\beta_{1|k}\beta_{n|k}}r_{n1} & \sqrt{\beta_{2|k}\beta_{n|k}}r_{n2} & \cdots & \beta_{n|k}r_{nn} \end{bmatrix}. \qquad (27)$$

where $n_{ij}$ and $r_{ij}$ are the elements of row $i$ and column $j$ of matrices $\mathbf{N}_k$ and $\mathbf{R}_k$, respectively.

When the $L_k$ or $L_{k-1}$ in Section 3.1 is smaller than the threshold $L_{\text{thr}}$, the possibility of NGN appearing in the observation is little or nothing. If the difference between $\mathbf{x}_{k|k-1}$ and $\mathbf{y}_k$ is still large under these condition, the motion state has changed, and $\mathbf{P}_{k|k-1}$ needs to be corrected to $\hat{\mathbf{P}}_{k|k-1}$.

Theoretically, $\hat{\mathbf{P}}_{k|k-1}$ needs to make Equation (28) valid. However, because $\mathbf{H}_k$ is not necessarily reversible, the value of $\hat{\mathbf{P}}_{k|k-1}$ cannot be directly calculated by Equation (28). Therefore, the expansion coefficient $\alpha_k$ is introduced to adjust $\hat{\mathbf{P}}_{k|k-1}$, as shown in Equations (29) and (30):

$$\tilde{\mathbf{C}}_k = \mathbf{H}_k\hat{\mathbf{P}}_{k|k-1}\mathbf{H}_k^{\mathrm{T}} + \mathbf{R}_k, \qquad (28)$$

$$\alpha_k = \max[1, \frac{\mathrm{tr}(\tilde{\mathbf{C}}_k - \mathbf{R}_k)}{\mathrm{tr}(\mathbf{H}_k\mathbf{P}_{k|k-1}\mathbf{H}_k^{\mathrm{T}})}], \qquad (29)$$

$$\hat{\mathbf{P}}_{k|k-1} = \alpha_k\mathbf{P}_{k|k-1}. \qquad (30)$$

Based on the above steps, a DST algorithm is proposed, as shown in Figure 3. Because there is a possibility of misjudgment when only $L_k$ is used, both $L_k$ and $L_{k-1}$ are required to be smaller than $L_{\text{thr}}$ to ensure the robustness of the algorithm.

The DST algorithm uses sensors information as a priori information to effectively adjust the noise covariance matrices. When the noise is non-stationary, the DST algorithm can correct the noise covariance matrix to be closer to the current noise in real time. This means that the DST algorithm can still ensure the accurate estimation of the noise covariance matrix in the presence of NGN. Combined with the DST algorithm, the robustness and accuracy of the MCKF will be improved significantly.

### 3.3. Dynamic Self-Tuning Maximum Correntropy Kalman Filter

The DST algorithm is combined with the MCKF, and then the dynamic self-tuning maximum correntropy Kalman filter (DSTMCKF) is proposed. The DST algorithm is used for correcting the values of $\mathbf{P}_{k|k-1}$ or $\mathbf{R}_k$ firstly. Then, the modified noise covariance matrices are used for the Kalman filter, and the filter result $\hat{\mathbf{x}}_{k|k}(0)$ is used as the starting point of the fixed-point iteration.

When the estimates of $\mathbf{P}_{k|k-1}$ and $\mathbf{R}_k$ are close to the true value, the value of $\hat{\mathbf{x}}_{k|k}(0)$ is very close to the maximum correntropy point. Taking $\hat{\mathbf{x}}_{k|k}(0)$ as the starting point of the fixed-point iteration ensures that there is only one maximum value in the solution space of the fixed-point iteration. Finally, the state quantity with the maximum correntropy is calculated by using the fixed-point iterative algorithm, which is the filtering result of DSTMCKF. The above process is expressed as Equations (31) to (48):

(a) Prediction:

$$\begin{cases} \hat{\mathbf{x}}_{k|k-1} = \mathbf{F}_k\hat{\mathbf{x}}_{k-1|k-1} \\ \mathbf{P}_{k|k-1} = \mathbf{F}_k\mathbf{P}_{k-1}\mathbf{F}_k^{\mathrm{T}} + \mathbf{Q}_{k-1} \end{cases}. \qquad (31)$$

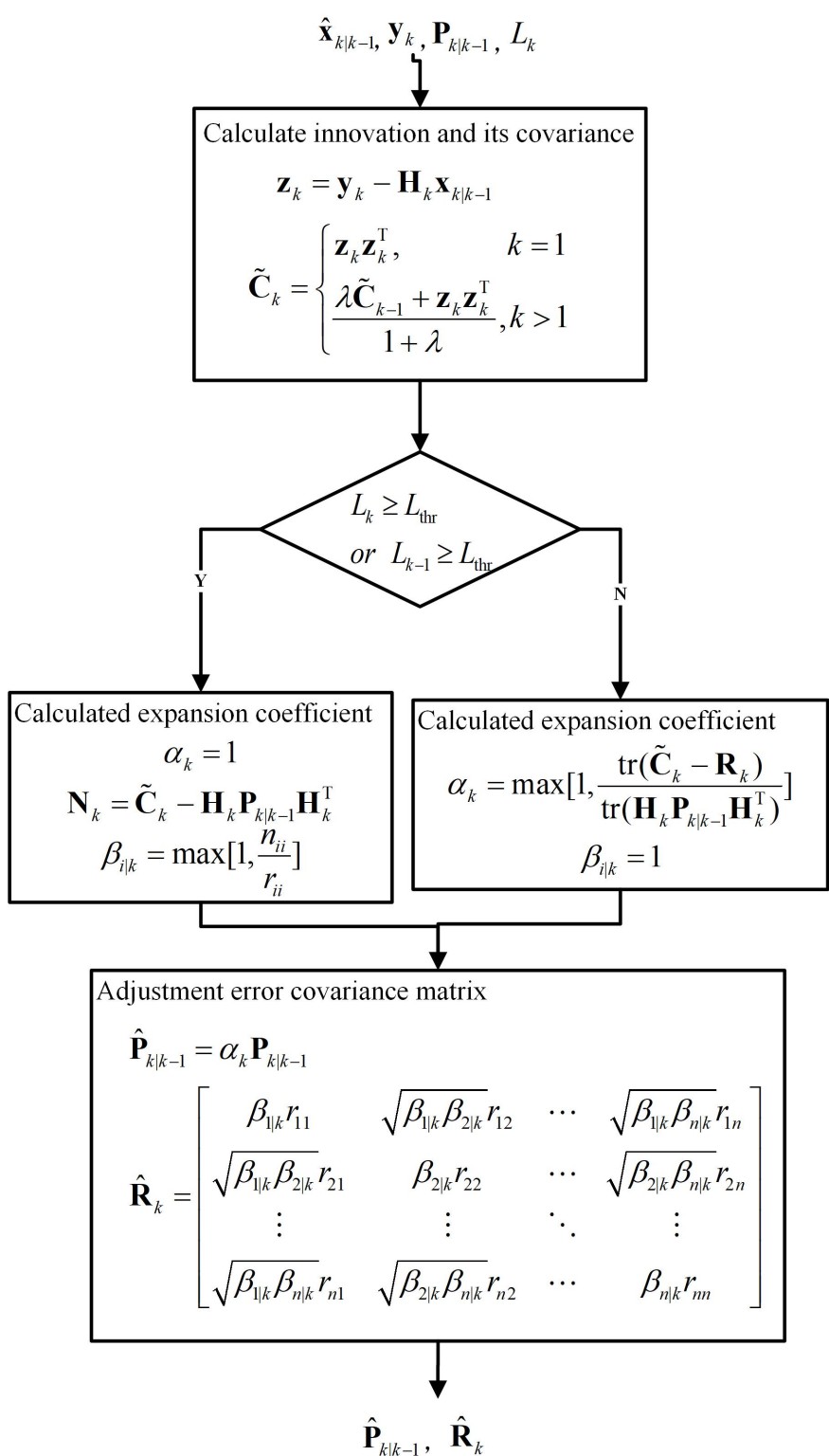

**Figure 3.** DST Algorithm.

(b)　Adjust noise covariance matrix by DST algorithm:

$$\mathbf{z}_k = \mathbf{y}_k - \mathbf{H}_k \mathbf{x}_{k|k-1},\qquad(32)$$

$$\tilde{\mathbf{C}}_k = \begin{cases} \mathbf{z}_k \mathbf{z}_k^{\mathrm{T}}, k=1 \\ \frac{\lambda \tilde{\mathbf{C}}_{k-1} + \mathbf{z}_k \mathbf{z}_k^{\mathrm{T}}}{1+\lambda}, k>1 \end{cases}.\qquad(33)$$

IF $L_k \geq L_{\text{thr}} | L_{k-1} \geq L_{\text{thr}}$:

$$\mathbf{N}_k = \tilde{\mathbf{C}}_k - \mathbf{H}_k \mathbf{P}_{k|k-1} \mathbf{H}_k^{\text{T}}, \tag{34}$$

$$\beta_{i|k} = \max[1, \frac{n_{ii}}{r_{ii}}], \tag{35}$$

$$\hat{\mathbf{P}}_{k|k-1} = \mathbf{P}_{k|k-1}, \tag{36}$$

$$\hat{\mathbf{R}}_k = \begin{bmatrix} \beta_{1|k} r_{11} & \sqrt{\beta_{1|k}\beta_{2|k}} r_{12} & \cdots & \sqrt{\beta_{1|k}\beta_{n|k}} r_{1n} \\ \sqrt{\beta_{1|k}\beta_{2|k}} r_{21} & \beta_{2|k} r_{22} & \cdots & \sqrt{\beta_{2|k}\beta_{n|k}} r_{2n} \\ \vdots & \vdots & \ddots & \vdots \\ \sqrt{\beta_{1|k}\beta_{n|k}} r_{n1} & \sqrt{\beta_{2|k}\beta_{n|k}} r_{n2} & \cdots & \beta_{n|k} r_{nn} \end{bmatrix}. \tag{37}$$

If $L_k < L_{\text{thr}}$ & $L_{k-1} < L_{\text{thr}}$:

$$\alpha_k = \max[1, \frac{\text{tr}(\tilde{\mathbf{C}}_k - \mathbf{R}_k)}{\text{tr}(\mathbf{H}_k \mathbf{P}_{k|k-1} \mathbf{H}_k^{\text{T}})}], \tag{38}$$

$$\hat{\mathbf{P}}_{k|k-1} = \alpha_k \mathbf{P}_{k|k-1}, \tag{39}$$

$$\hat{\mathbf{R}}_k = \mathbf{R}_k. \tag{40}$$

(c) Calculation of fixed-point iteration starting point by KF:

$$\mathbf{K}_k = \hat{\mathbf{P}}_{k|k-1} \mathbf{H}_k^{\text{T}} [\mathbf{H}_k \hat{\mathbf{P}}_{k|k-1} \mathbf{H}_k^{\text{T}} + \hat{\mathbf{R}}_k]^{-1}, \tag{41}$$

$$\hat{\mathbf{x}}_{k|k}(0) = \hat{\mathbf{x}}_{k|k-1} + \mathbf{K}_k [\mathbf{y}_k - \mathbf{H}_k \hat{\mathbf{x}}_{k|k-1}]. \tag{42}$$

(d) Parameters required to calculate MCKF:

$$\mathbf{B}_p = \text{chol}(\hat{\mathbf{P}}_{k|k-1}), \tag{43}$$

$$\mathbf{B}_r = \text{chol}(\hat{\mathbf{R}}_k), \tag{44}$$

$$\mathbf{B}_k = \begin{bmatrix} \mathbf{B}_p & 0 \\ 0 & \mathbf{B}_r \end{bmatrix}, \tag{45}$$

$$\mathbf{D}_k = \mathbf{B}_k^{-1} [\ \hat{\mathbf{x}}_{k|k-1} \quad \mathbf{y}_k \ ]^{\text{T}}, \tag{46}$$

$$\mathbf{W}_k = \mathbf{B}_k^{-1} [\ \mathbf{I} \quad \mathbf{H}_k \ ]^{\text{T}}, \tag{47}$$

$$\mathbf{E}_k = \mathbf{D}_k - \mathbf{W}_k \hat{\mathbf{x}}_{k|k}. \tag{48}$$

(e) Fixed-point iteration:

$$\begin{cases} \tilde{\mathbf{C}}_x = \text{diag}(G_\sigma(\tilde{\mathbf{e}}_k(1)), \cdots, G_\sigma(\tilde{\mathbf{e}}_k(n))) \\ \tilde{\mathbf{C}}_y = \text{diag}(G_\sigma(\tilde{\mathbf{e}}_k(n+1)), \cdots, G_\sigma(\tilde{\mathbf{e}}_k(n+m))) \end{cases}, \tag{49}$$

$$\begin{cases} \tilde{\mathbf{R}}_k = \mathbf{B}_r \tilde{\mathbf{C}}_y^{-1} \mathbf{B}_r^T \\ \tilde{\mathbf{P}}_{k|k-1} = \mathbf{B}_p \tilde{\mathbf{C}}_x^{-1} \mathbf{B}_p^T \end{cases}, \tag{50}$$

$$\tilde{\mathbf{K}}_k = \tilde{\mathbf{P}}_{k|k-1} \mathbf{H}_k^{\text{T}} [\mathbf{H}_k \tilde{\mathbf{P}}_{k|k-1} \mathbf{H}_k^{\text{T}} + \tilde{\mathbf{R}}_k]^{-1}, \tag{51}$$

$$\hat{\mathbf{x}}_{k|k}(t) = \hat{\mathbf{x}}_{k|k-1} + \tilde{\mathbf{K}}_k [\mathbf{y}_k - \mathbf{H}_k \hat{\mathbf{x}}_{k|k-1}]. \tag{52}$$

(f) When Equation (53) holds, the fixed-point iteration is stopped; otherwise, continue with step (e):

$$\frac{\left\| \hat{\mathbf{x}}_{k|k}(t) - \hat{\mathbf{x}}_{k|k}(t-1) \right\|}{\left\| \hat{\mathbf{x}}_{k|k}(t) \right\|} \leq \varepsilon. \tag{53}$$

According to the above steps, the filtering result of DSTMCKF is obtained. The process of DSTMCKF is also summarized as shown in Figure 4, and DST algorithm is in Figure 3.

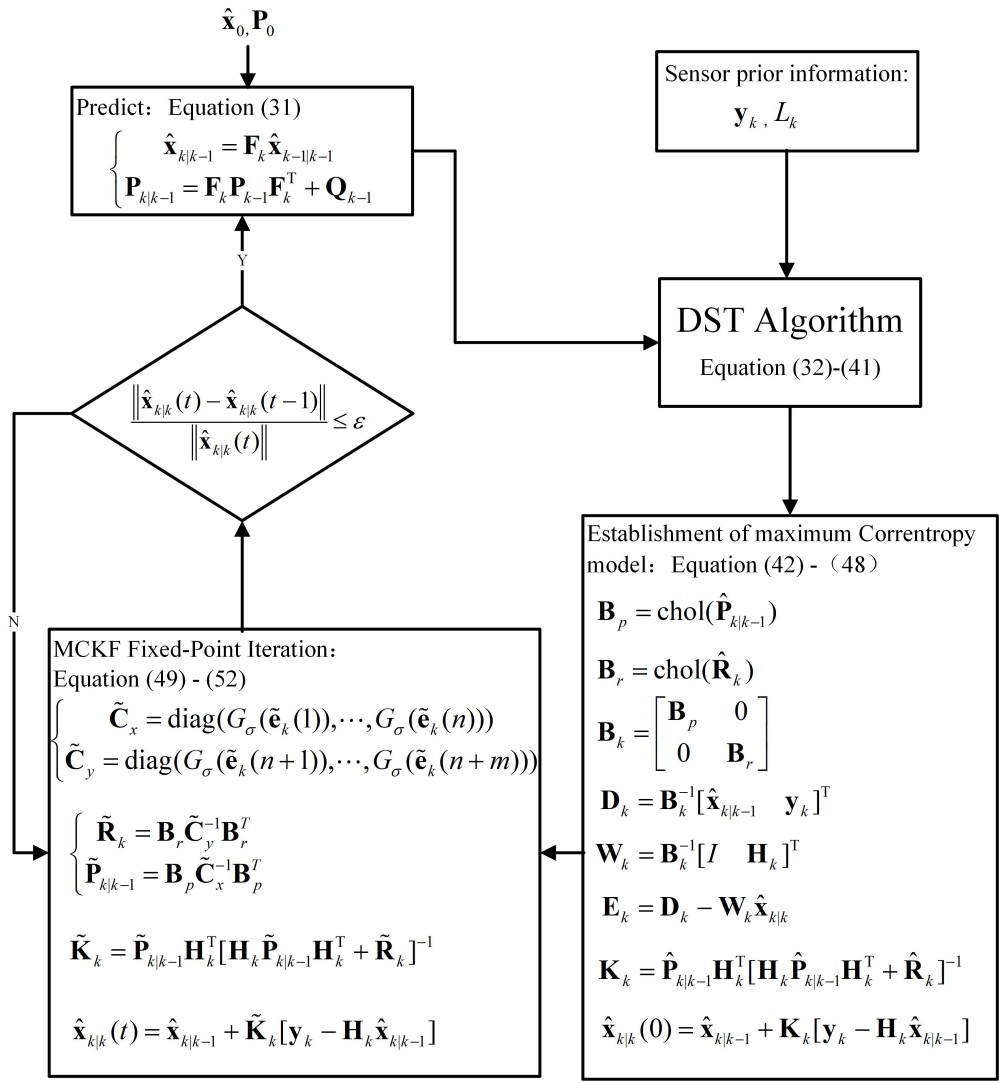

**Figure 4.** Flowchart of DSTMCKF algorithm.

When combined with the DST algorithm, the ability of the MCKF to process NGN is theoretically improved. The noise covariance matrices, which can be dynamically corrected with the noise size, enable the DSTMCKF to have higher accuracy and robustness than the MCKF. Compared with the VKW-MCKF, the DSTMCKF still works well when the noise covariance matrices estimation is inaccurate. The VKW-MCKF, which can only adjust the size of the Gaussian kernel bandwidth, is theoretically less efficient than the DSTMCKF when estimating the noise covariance matrices inaccurately. In addition, compared with the R-MEEKF, the DSTMCKF does not need to balance accuracy with the response speed and has stronger robustness in theory.

## 4. Simulation and Application in Office Indoor Environment

### 4.1. Simulation of Indoor Positioning Scene and Analysis of Filtering Results

In order to verify the performance of the DSTMCKF in WSNs positioning systems, the shared situations in the WSNs positioning systems are considered for simulation firstly. Two motion states and three kinds of noise between the anchors and tag are considered.

The simulation sampling frequency is set to 100 Hz. The two motion states are the uniform linear motion process and the acceleration–deceleration process. During the uniform linear motion process, the speed of the tag in the $x$-direction is 1 m/s, the speed in the $y$-direction is 0 m/s and the process noise on the speed in each sampling is $\omega_k \sim N(0, 0.01)$. During the acceleration–deceleration process, the acceleration of the tag in the $x$-direction is zero, the acceleration in the $y$-direction is $-5$ m/s$^2$ in the first 0.5 s and 5 m/s$^2$ in the last 0.5 s. The process noise on the speed in each sampling is $\omega_k \sim N(0, 0.025)$.

The simulation environment is shown in Figure 5. The simulation uses three anchors with coordinates of (0,0), (20,0) and (10,17.32) to form an equilateral triangle with a side length of 20 m. There are three kinds of shared noises between the anchors and tag. Note that the three kinds of noise are $S_1$, $S_2$ and $S_3$, and the distribution of the noise is shown in Table 1, where $A_i$ represents the channel between the $i$-th anchor and the tag.

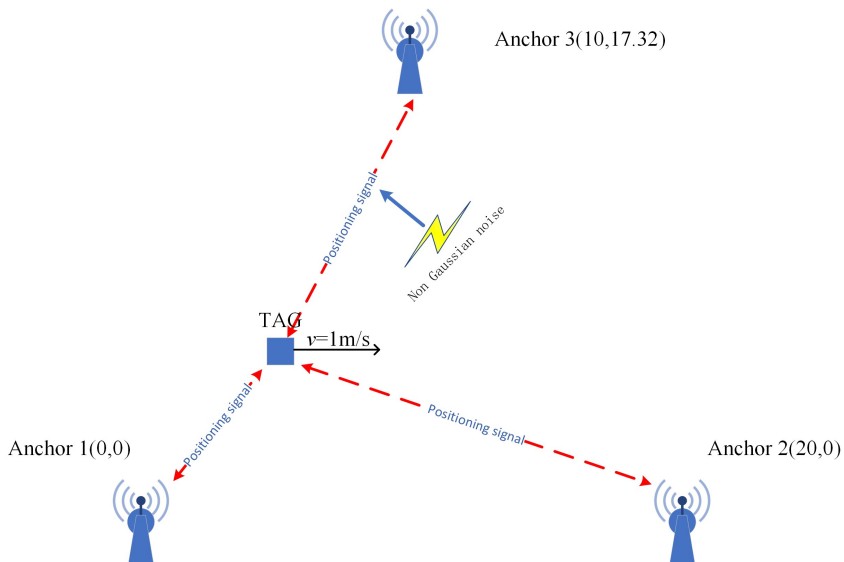

**Figure 5.** Simulated environment.

**Table 1.** Three kinds of noise between the tag and the anchors.

| Noise | Distribution (m) | Channel with Noise |
|:---:|:---:|:---:|
| $S_1$ | $S_k \sim N(0, 0.02)$ | $A_1, A_2, A_3$ |
| $S_2$ | $lnS_k \sim N(0, 0.05)$ | $A_1, A_2, A_3$ |
| $S_3$ | $S_k \sim U(0, 0.3) + |N(0, 0.2)|$ | $A_3$ |

Under the above simulation conditions, the MCKF, VKW-MCKF, R-MEEKF and DSTM-CKF are used for the comparative experiments. Assuming that there is only noise $S_1$, the significance level of the hypothesis testing is taken as 0.05, and then the threshold value $L_{\text{thr}}$ is calculated as 0.13 . The state transition matrix $\mathbf{F}_k$ is obtained according to the kinematic characteristics of the simulation object, as shown in Equation (54). Because the WSNs can only measure the position of the simulation object, the observation matrix $\mathbf{H}_k$ is shown in Equation (55).

$$\mathbf{F}_k = \begin{bmatrix} 1 & T & 0 & 0 \\ 0 & 1 & 0 & 0 \\ 0 & 0 & 1 & T \\ 0 & 0 & 0 & 1 \end{bmatrix}, \tag{54}$$

$$\mathbf{H}_k = \begin{bmatrix} 1 & 0 & 0 & 0 \\ 0 & 0 & 1 & 0 \end{bmatrix}. \tag{55}$$

Four cases are discussed in the simulation experiment, as shown below:

Case (1): In Case (1), there is an acceleration–deceleration phase and noise $S_1$ in Table 1. The Gaussian kernel $\sigma$ of the MCKF, R-MEEKF and DSTMCKF is 1.2, and because the VKW-MCKF will diverge when the Gaussian kernel $\sigma$ is 1.2, the Gaussian kernel $\sigma$ of the VKW-MCKF in the figure is 5.0. The simulation sampling results and experimental results are shown in Figure 6, where Figure 6e is the cumulative distribution function (CDF) of each filter.

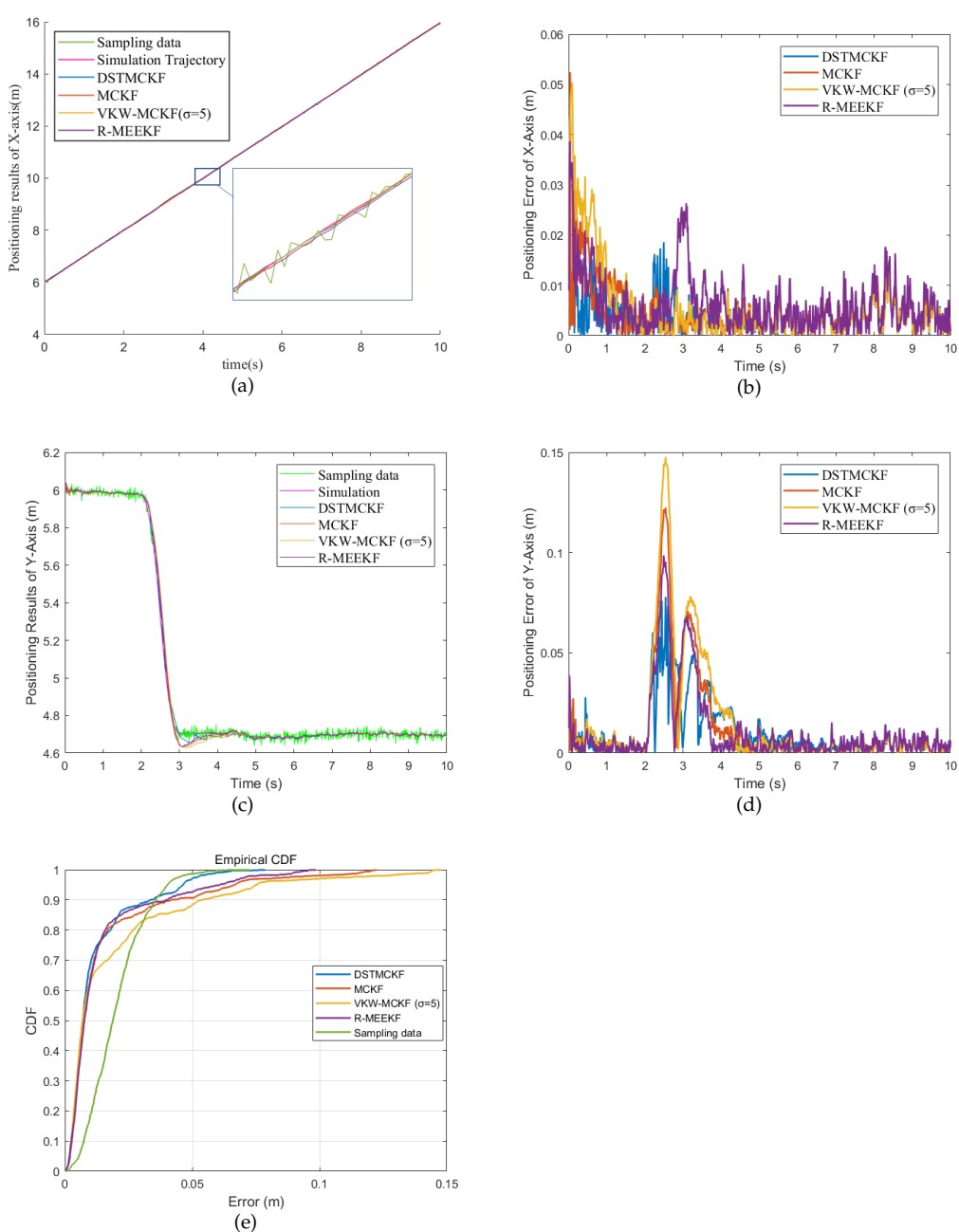

**Figure 6.** Simulation Results in Case (1). (**a**) Positioning result of X-label when $\sigma = 1.2$; (**b**) error of X-label when $\sigma = 1.2$; (**c**) positioning result of Y-label when $\sigma = 1.2$; (**d**) error of Y-label when $\sigma = 1.2$; (**e**) CDF of each filter when $\sigma = 1.2$.

The experimental results indicate that when the motion states are multiple, the DSTM-CKF has the highest accuracy, while the filtering accuracy of the MCKF has decreased to a certain extent. In order to discuss the influence of bandwidth $\sigma$ on the filtering results, the value of $\sigma$ is modified, and the results are shown in Table 2.

When the value of $\sigma$ is small, the dynamic tracking ability of the MCKF is underdeveloped. In addition, the error is large or even divergent when the motion state is multiple. With the increase of $\sigma$, the MCKF gradually converges to KF, and the dynamic performance is improved.

The response speed of the VKW-MCKF and R-MEEKF is improved to some extent compared with the MCKF, so the positioning accuracy is higher in the presence of multiple motion states. However, because neither of the two algorithms have the ability to adjust the noise covariance values adaptively, the accuracy decreases when the observed noise does not match the estimation.

For the DSTMCKF, a small $\sigma$ also leads to a decrease in the dynamic performance, and an increase of $\sigma$ will invalidate the MCC. Thus, the DSTMCKF will perform best only when $\sigma$ is moderate. Nonetheless, the dynamic tuning of parameters reduces the sensitivity of the DSTMCKF to $\sigma$, and the RMSE of the DSTMCKF changes little when $\sigma$ changes.

**Table 2.** RMSE (m) with different $\sigma$ (Case 1).

| $\sigma$ | DSTMCKF | MCKF | VKW-MCKF | R-MEEKF |
|---|---|---|---|---|
| 0.4 | 0.021 | 0.055 | Divergence | 0.029 |
| 0.8 | 0.019 | 0.028 | Divergence | 0.023 |
| 1.2 | 0.018 | 0.027 | Divergence | 0.023 |
| 2.0 | 0.016 | 0.026 | Divergence | 0.023 |
| 5.0 | 0.015 | 0.026 | 0.033 | 0.023 |
| 10.0 | 0.015 | 0.026 | 0.032 | 0.024 |
| 20.0 | 0.015 | 0.026 | 0.032 | 0.025 |

Case (2): In Case (2), the noise $S_1$ in Table 1 occurs in the whole process and the noise $S_2$ in Table 1 occurs during iterations 500 to 600 s. The Gaussian kernel $\sigma$ of the MCKF, R-MEEKF and DSTMCKF is 1.2. In addition, because the VKW-MCKF will diverge when the Gaussian kernel $\sigma$ is 1.2, the Gaussian kernel $\sigma$ of the VKW-MCKF in the figure is 5.0. The simulation sampling results and experimental results are shown in Figure 7.

The DSTMCKF performs better than the MCKF when $\sigma$ is 1.2. However, because the motion state is single at this time, $\sigma$ takes a smaller value to improve the performance of the MCKF. The experimental results are shown in Table 3. When $\sigma$ is small enough, the accuracy of the MCKF is improved to a certain extent. Nonetheless, to ensure the dynamic performance of the filter, it is difficult for $\sigma$ to obtain such a small value.

**Table 3.** RMSE (m) with different $\sigma$ (Case 2).

| $\sigma$ | DSTMCKF | MCKF | VKW-MCKF | R-MEEKF |
|---|---|---|---|---|
| 0.4 | 0.007 | 0.019 | Divergence | 0.032 |
| 0.8 | 0.007 | 0.041 | Divergence | 0.035 |
| 1.2 | 0.007 | 0.043 | Divergence | 0.042 |
| 2.0 | 0.008 | 0.044 | Divergence | 0.046 |
| 5.0 | 0.009 | 0.045 | 0.013 | 0.047 |
| 10.0 | 0.009 | 0.045 | 0.019 | 0.047 |
| 20.0 | 0.010 | 0.045 | 0.030 | 0.047 |

Due to the variable bandwidth of the Gaussian kernel, the VKW-MCKF has a better ability to deal with NGN in observation, and the positioning accuracy is almost the same as that of the DST-MCKF. The steady-state error of the R-MEEKF is lower than that of the MCKF, but the corresponding convergence speed is slower, and the anti-jamming ability is underdeveloped; therefore, the positioning accuracy of the R-MEEKF is not better than

the MCKF. The DSTMCKF has the ability to adjust the noise covariance matrices during operation; hence, it has a better ability to handle the NGN in the observations and has the highest accuracy.

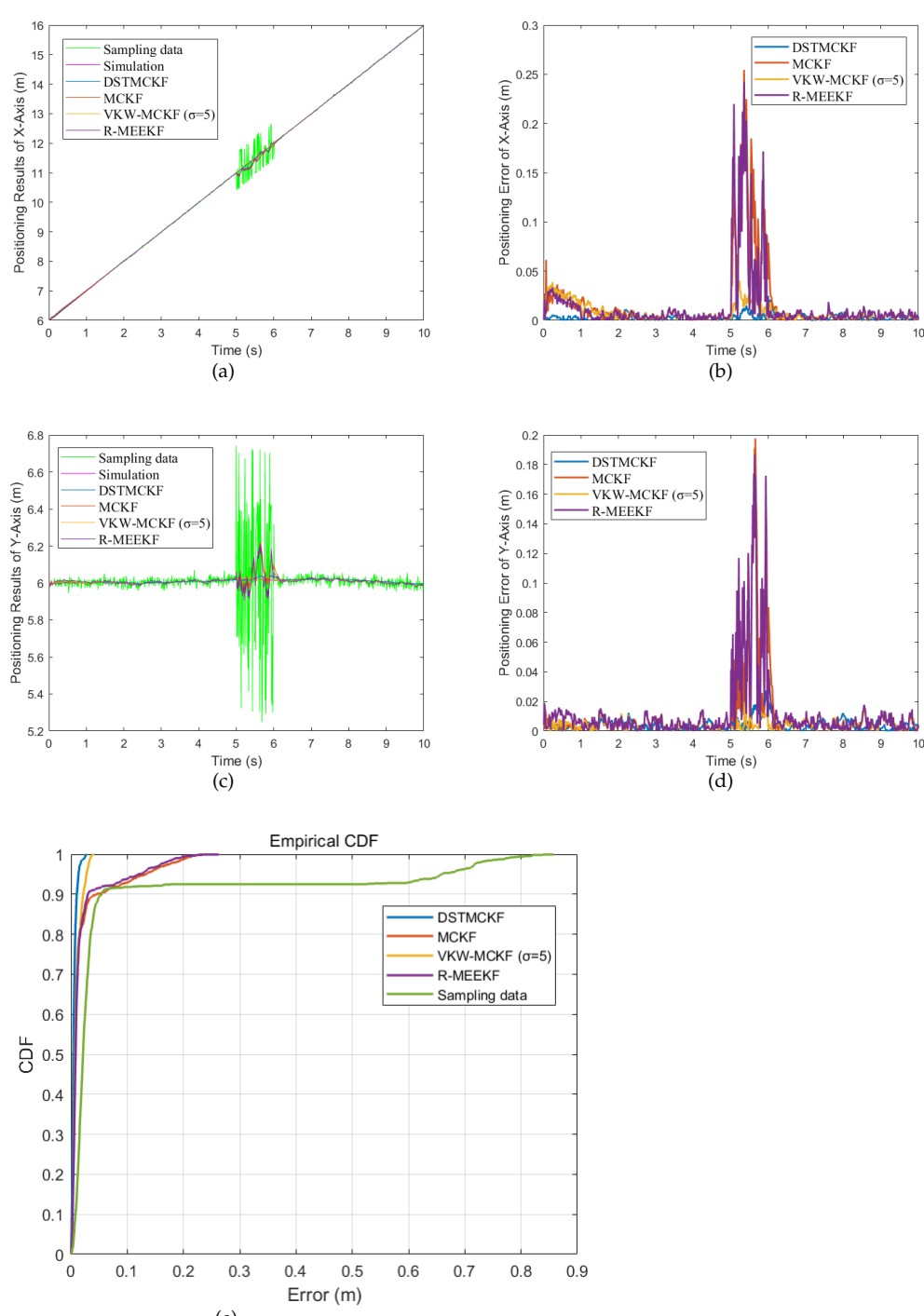

**Figure 7.** Simulation Results in Case (2). (**a**) Positioning result of X-label when $\sigma = 1.2$; (**b**) error of X-label when $\sigma = 1.2$; (**c**) positioning result of Y-label when $\sigma = 1.2$; (**d**) error of Y-label when $\sigma = 1.2$; (**e**) CDF of each filter when $\sigma = 1.2$.

Case (3): In Case (3), the noise $S_1$ in Table 1 occurs in the whole process and the noise $S_3$ in Table 1 occurs during iterations 700 to 800. The Gaussian kernel $\sigma$ of the MCKF, R-MEEKF and DSTMCKF is 1.2. However, because the VKW-MCKF will diverge when the

Gaussian kernel $\sigma$ is 1.2, the Gaussian kernel $\sigma$ of the VKW-MCKF in the figure is 5.0. The simulation sampling results and experimental results are shown in Figure 8.

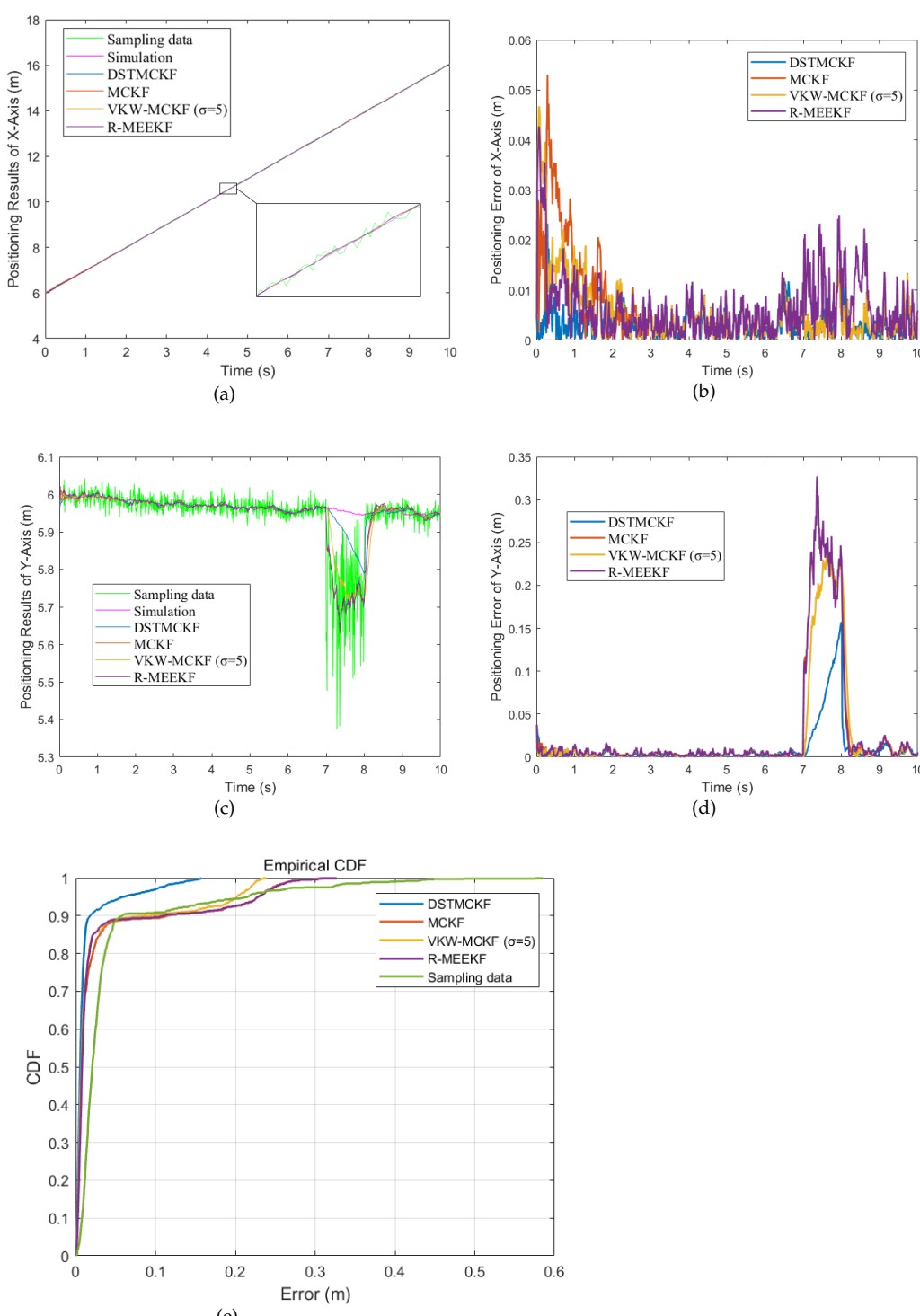

**Figure 8.** Simulation Results in Case (3). (**a**) Positioning result of X-label when $\sigma = 1.2$; (**b**) error of X-label when $\sigma = 1.2$; (**c**) positioning result of Y-label when $\sigma = 1.2$; (**d**) error of Y-label when $\sigma = 1.2$; (**e**) CDF of each filter when $\sigma = 1.2$.

It can be seen from the figure that in case (3), the accuracy of the MCKF is also lower than that of the DSTMCKF. And not similar to case (2), the performance of the MCKF is not improved by reducing the value of $\sigma$, as shown in Table 4.

**Table 4.** RMSE (m) with different $\sigma$ (Case 3).

| $\sigma$ | DSTMCKF | MCKF | VKW-MCKF | R-MEEKF |
|---|---|---|---|---|
| 0.4 | 0.008 | 0.073 | Divergence | 0.072 |
| 0.8 | 0.023 | 0.073 | Divergence | 0.072 |
| 1.2 | 0.027 | 0.074 | Divergence | 0.073 |
| 2.0 | 0.032 | 0.074 | Divergence | 0.074 |
| 5.0 | 0.034 | 0.074 | 0.063 | 0.074 |
| 10.0 | 0.034 | 0.074 | 0.069 | 0.074 |
| 20.0 | 0.034 | 0.074 | 0.072 | 0.074 |

In Table 4, even if $\sigma$ is small enough, the MCKF will be severely affected by the noise. Conversely, the DSTMCKF still ensures a high accuracy no matter how the value of $\sigma$ changes. This means that the DSTMCKF will still converge when the motion state changes and has higher robustness and numerical stability than the MCKF. This means that the DSTMCKF has a stronger performance in processing NGN with offset.

The VKW-MCKF has a high accuracy when the Gauss kernel bandwidth $\sigma$ is small, yet the numerical stability of the VKW-MCKF is underdeveloped. In practice, the Gauss kernel bandwidth $\sigma$ is usually larger. The R-MEEKF is relatively insensitive to the change of the Gaussian kernel $\sigma$ and has strong numerical stability, but the accuracy of the R-MEEKF is the lowest.

Case (4): In Case (4), there is an acceleration–deceleration phase in the motion state. The noise $S_1$ in Table 1 occurs in the whole process, the noise $S_2$ in Table 1 occurs during iterations 500 to 600 and the noise $S_3$ in Table 1 occurs during iterations 700 to 800. The Gaussian kernel $\sigma$ of the MCKF, R-MEEKF and DSTMCKF is 1.2. In addition, because the VKW-MCKF will diverge when the Gaussian kernel $\sigma$ is 1.2, the Gaussian kernel $\sigma$ of the VKW-MCKF in the figure is 5.0. The simulation sampling results and experimental results are shown in Figure 9.

When the motion state and noise type are multiple, it is difficult for $\sigma$ to obtain an appropriate value to make the MCKF have better accuracy. In order to discuss the influence of bandwidth $\sigma$ on the filtering results, the value of $\sigma$ is modified, and the results are shown in Table 5.

**Table 5.** RMSE (m) with different $\sigma$ (Case 4).

| $\sigma$ | DSTMCKF | MCKF | VKW-MCKF | R-MEEKF |
|---|---|---|---|---|
| 0.4 | 0.042 | 0.104 | Divergence | 0.094 |
| 0.8 | 0.044 | 0.102 | Divergence | 0.100 |
| 1.2 | 0.046 | 0.102 | Divergence | 0.105 |
| 2.0 | 0.048 | 0.101 | Divergence | 0.103 |
| 5.0 | 0.048 | 0.103 | 0.079 | 0.104 |
| 10.0 | 0.048 | 0.103 | 0.086 | 0.105 |
| 20.0 | 0.051 | 0.104 | 0.101 | 0.105 |

When the value of $\sigma$ is small, the dynamic tracking ability of the MCKF is underdeveloped. On the contrary, when the value of $\sigma$ increases, the ability of the MCKF to deal with NGN decreases. Therefore, when $\sigma$ is equal to 2.0, the MCKF has the best accuracy. Nevertheless, the DSTMCKF has the ability to self-tune parameters, and it has the highest accuracy no matter how $\sigma$ changes. The DSTMCKF also performs better in a simulation than the VKW-MCKF and R-MEEKF.

In addition, in each iteration, the averaged simulation time for the DSTMCK, MCKF and Kalman filter is 0.144, 0.096 and 0.025 ms, respectively. For the control systems, in order to meet the control requirements, it is required that the frequency of the uploading positioning results can reach more than 100 Hz [31]. Although the DSTMCKF is more computationally burdensome, it is still within the acceptable range.

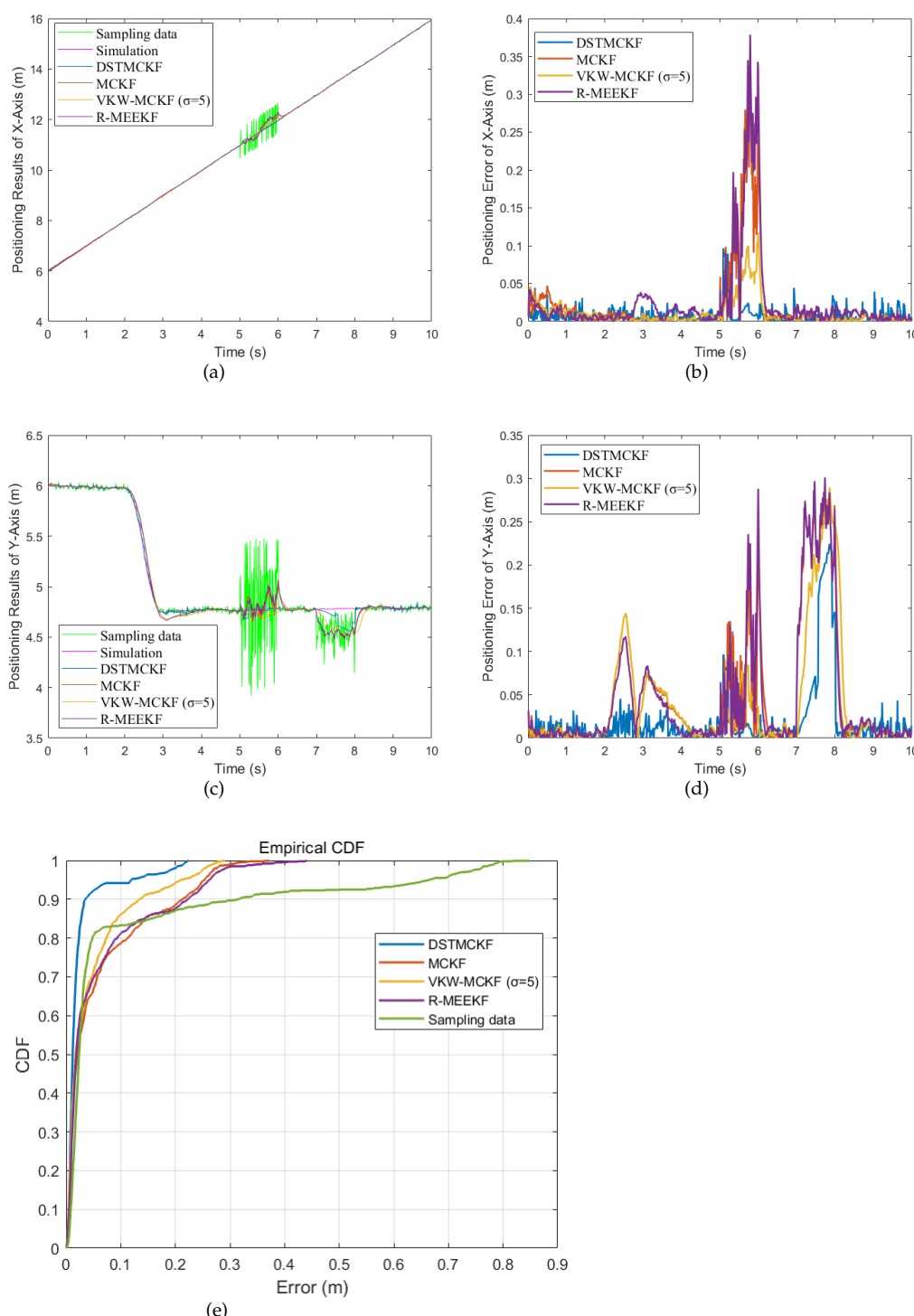

**Figure 9.** Simulation Results in Case (4). (**a**) Positioning result of X-label when $\sigma = 1.2$; (**b**) error of X-label when $\sigma = 1.2$; (**c**) positioning result of Y-label when $\sigma = 1.2$; (**d**) error of Y-label when $\sigma = 1.2$; (**e**) CDF of each filter when $\sigma = 1.2$.

In summary, the DSTMCKF outperforms the MCKF and its improvement method in all four cases and effectively handles situations where there is no single motion state or noise type. Meanwhile, the influence of $\sigma$ on the DSTMCKF is limited, which indicates that the DSTMCKF has better numerical stability and robustness than the MCKF. The simulation results show that the DSTMCKF is more suitable for WSNs positioning systems than the MCKF and its improvement method.

### 4.2. Application in Office Indoor Environment

In order to prove that the DSTMCKF is applied to the actual WSNs positioning systems, experiments are carried out with an ultra-wideband (UWB) positioning network in the real environment, and the experimental environment in the real world is shown in Figure 10. The indoor positioning environment selected in this experiment is a common office environment, with an area of about $10 \times 7.5 \text{ m}^2$. In the experiment, three UWB anchors and one UWB tag are selected to form a UWB positioning network. The positions of the three anchors are shown in Figure 11a, respectively, (0, 0), (8.9, 0) and (4.45, 7.4). The tag is placed on the robot and travels around the indoor aisle. There are several mark points in the room, and the robot counts the time when passing through the mark points. The recorded time and the coordinates of the mark points are used as features for Gaussian smoothing to obtain an ideal path. The ideal path is shown by the orange line in the figure, and the sampled data are shown by the blue line in the figure.

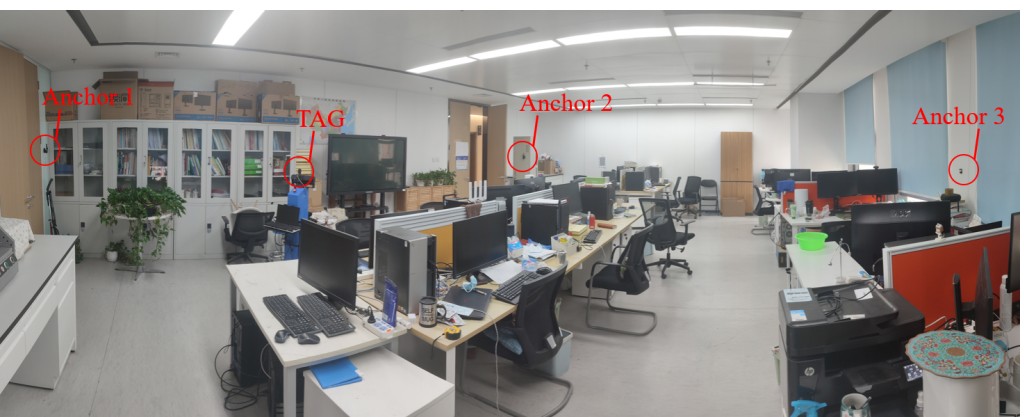

**Figure 10.** Experimental environment in real world.

The positioning result of the UWB is affected by various NGNs. Most of these NGNs come from the electromagnetic radiation of electronic equipment and non-line-of-sight signals due to occlusion and refraction. To deal with these NGNs, the VKW-MCKF, R-MEEKF, MCKF and DSTMCKF are used in the experiment, and the results are shown in Figure 11b–d.

It is seen from the figure that when NGN does not exist, the performance of the three filters is almost the same. When NGN exists, the accuracy and robustness of the DSTMCKF are better than the MCKF, VKW-MCKF and R-MEEKF. The RMSE and maximum error (ME) of the three filtering results are shown in the Table 6. The experiment results indicate that the DSTMCKF also has better performance than the MCKF and its improvement method, and the error is reduced by 34.5% compared to the MCKF.

**Table 6.** RMSE and maximum error of three filters.

|  | RMSE (m) | ME (m) |
|---|---|---|
| DSTMCKF | 0.036 | 0.17 |
| MCKF | 0.055 | 0.18 |
| VKW-MCKF | 0.063 | 0.33 |
| R-MEEKF | 0.060 | 0.35 |

To sum up, the DSTMCKF solves the problem of the underdeveloped performance of the MCKF in the noise type or motion state to a certain extent. It has a higher accuracy and robustness and significantly improves the positioning accuracy of the systems in the actual WSNs positioning systems.

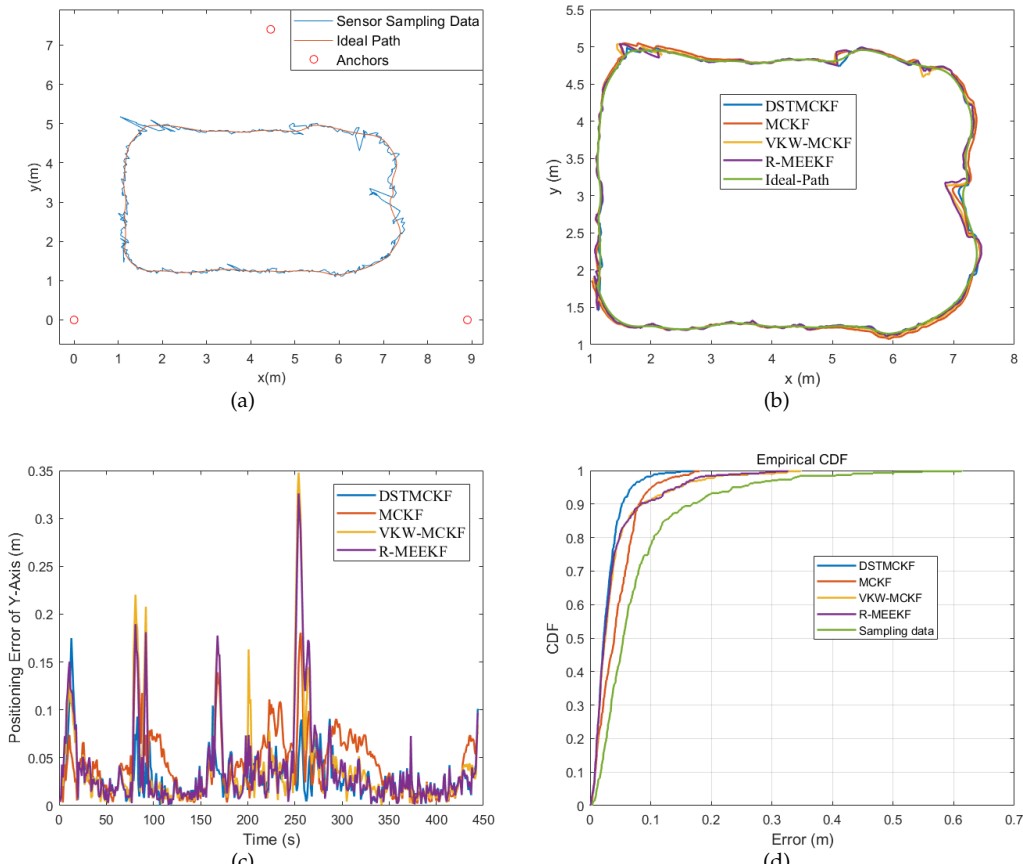

**Figure 11.** Results in real world. (**a**) Sampling results in real world; (**b**) filtering results in real world; (**c**) filtering error in real world; (**d**) CDF of each filter.

## 5. Discussion

The range-based positioning method is the most shared positioning method in WSNs positioning systems, and the positioning results are affected by NGN [32–36]. To improve the accuracy of WSNs positioning systems, a DSTMCKF is proposed. Four simulation scenarios are considered to test the performance of the DSTMCKF, and the DSTMCKF has a significantly improved performance in various scenarios compared to other common filtering algorithms. To further prove the reliability of the DSTMCKF, the algorithm is used in real-world indoor scenes. The results show that the DSTMCKF has a higher accuracy than other filtering algorithms, can adapt to a complex noise environment in real scenes and effectively improves the accuracy of a WSNs positioning system.

The main idea of this paper is to improve the accuracy of a WSNs positioning system with a DSTMCKF. We mainly discuss the effects of the non-Gaussian process and observation noise on the positioning results and use the DSTMCKF to mitigate these effects. Because the NGN in the real-world scene is not as extreme as the NGN in the simulation scene, the advantages of the DSTMCKF are not as obvious as those in the simulation experiments. If the positioning experiments can be conducted in more complex indoor scenes, the advantages of the DSTMCKF will be further highlighted. In future work, we plan to investigate the improvements and extensions of the method for determining the type of noise. This helps reduce the risk of filter divergence due to misidentified noise types, improves filter accuracy in complex environments and enhances the robustness of the filter.

## 6. Conclusions

A dynamic self-tuning maximum correntropy Kalman filter is proposed to improve the performance of an MCKF. The DSTMCKF realizes the dynamic self-tuning of noise covariance matrices on the basis of innovation and a priori information from sensors. Applying the adjusted noise covariance matrices to the MCC effectively improves the performance of the MCKF when there is NGN or a motion state that does not conform to the estimate. In this way, the DSTMCKF solves the problem that the MCKF performs underdeveloped when there is NGN or a motion state that does not conform to the estimate. The simulation and experiments in the real-world environment indicate that the DSTMCKF also has a better performance than the MCKF, and the error of the DSTMCKF is reduced by 34.5, 42.9 and 40.0%, respectively, compared with the MCKF, VKW-MCKF and R-MEEKF in the real-world environment.

The experimental results prove that the DSTMCKF improved the response speed to NGN and the accuracy by using the DST algorithm to adjust the noise covariance matrices in real time. It also proves that the DSTMCKF improved the ability to process non-Gaussian noise by combining the DST algorithm with the MCKF.

The proposed DSTMCKF improves the filter precision in a WSNs positioning system, which improves the positioning accuracy of the WSNs positioning system. A higher positioning accuracy brings more scenarios for WSNs positioning systems. This means that WSNs positioning systems using a DSTMCKF are qualified for use in scenarios with higher accuracy requirements, such as automatic parking.

**Author Contributions:** This paper is a collaborative work by all the authors. T.L. proposed the main idea, designed the experiments, performed the experiments, analyzed the data and wrote the manuscript. K.H. and Y.D. added some ideas, gave suggestions and revised the rough draft. X.W. and S.S. assisted with certain experiments, and all authors proofread the paper. All authors have read and agreed to the published version of the manuscript.

**Funding:** This research was sponsored by the Beijing Municipal Natural Science Foundation Committee (L191020) and the China National Railway Corporation (P2021T002).

**Data Availability Statement:** Not applicable.

**Acknowledgments:** We would like to acknowledge the School of Automation, Beijing Institute of Technology, for the support in our research.

**Conflicts of Interest:** The authors declare no conflict of interest.

## Abbreviations

The following abbreviations are used in this manuscript:

| | |
|---|---|
| WSNs | Wireless Sensors Networks |
| MCKF | Maximum Correntropy Kalman Filter |
| DSTMCKF | Dynamic Self-Tuning Maximum Correntropy Kalman Filter |
| VKW-MCKF | Variable Kernel Width–Maximum Correntropy Kalman Filter |
| R-MEEKF | Robust Minimum Error Entropy Kalman Filter |
| DST | Dynamic Self Tuning |
| NGN | Non-Gaussian noise |
| MMSE | Minimum Mean Squared Error |
| CDF | Cumulative Distribution Function |
| MCC | Maximum Correntropy Criterion |
| RMSE | Root Mean Squared Error |
| ME | Maximum Error |

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
