# Peer review of "A Dynamic Self-Tuning Maximum Correntropy Kalman Filter for Wireless Sensors Networks Positioning Systems"

_remotesensing, doi:10.3390/rs14174345_

Round 1
Reviewer 1 Report
The paper deals with Kalman filtering in presence of non-Gaussian noises. Authors consider the “Maximum Correntropy Kalman Filter » (MCKF) and propose to enrich it by adding a “dynamic self-tuning”, resulting in the new “Dynamic Self-Tuning MCKF” (DSTMCKF). Thus DSTMCKF is a kind of adaptive MCKF. There is no theoretical contribution and the interest of the DSTMCKF is shown through experiments on simulated and real data. This experimental efficiency makes the main interest of the paper.
In the whole, the ideas of the paper are interesting and contributions appear as new enough to deserve publication. However, under its current form, the paper is rather confused and gives the impression of being written in a somewhat rushed way. It should be improved before being considered for publication. Some suggestions are specified below.
Some general suggestions are:
(i) the principle of maximum correntropy should be recalled before formulas (4)-(15) giving the solution in MCKF;
(ii) why following this principle is likely to improve the classic KF when the noise is not Gaussian should be specified. Is there any theoretical justification? In addition, NGN are mentioned through the whole paper, but Gaussian noises are considered in experiments … Please clarify this point;
(iii) DST algorithm described on page 7 seems to be well suited to deal with non stationarity; is it possibly the deep reason of its efficiency?
(iv) there are many English typos or inaccuracies. The paper should be carefully corrected by a native English reader.
In non-exhaustive manner, some specific suggestions are:
-p1, l26: what you mean by “multiple” noise or state? This is the core point, but you don’t define the word;
- p1, l32: “Maximum Correntropy Criterion” is not an “optimal criterion for filtering”, it is a “criterion for filter’s optimality”;
-p2, l55: specify what “MCC” means
-p2 the sequence (4)-(15) of equations is confusing. For example, P(k/k-1) appearing in (5) is specified in (10). What is more, it depends on P(k), which is not defined at all … same for different x.
p3, lines 64, 67: “a priori estimate” or “priori estimate”? Why “priori”? Explain. Similarly, p. 4, l.78, and Figures 1, 2 : why “prior” estimate?
…
Reviewer 2 Report
no comments
Reviewer 3 Report
Please find my remarks in the attached file.

Author Response
请参阅附件

Round 2
Reviewer 1 Report
The authors took my remarks correctly into accountReviewer 3 Report
My remarks have been carefully addressed, therefore I have no further comments.